# Worldwide Surveillance Actions and Initiatives of Drinking Water Quality: A Scoping Review

**DOI:** 10.3390/ijerph20010559

**Published:** 2022-12-29

**Authors:** Rayssa Horacio Lopes, Cícera Renata Diniz Vieira Silva, Ísis de Siqueira Silva, Pétala Tuani Cândido de Oliveira Salvador, Léo Heller, Severina Alice da Costa Uchôa

**Affiliations:** 1Graduation Program in Collective Health, Federal University of Rio Grande do Norte, Natal 59064-630, Brazil; 2Technical School of Health of Cajazeiras, Federal University of Campina Grande, Cajazeiras 58900-000, Brazil; 3School of Health, Federal University of Rio Grande do Norte, Natal 59078-970, Brazil; 4René Rachou Institute, Oswaldo Cruz Foundation, Belo Horizonte 30190-009, Brazil; 5Public Health Department, Federal University of Rio Grande do Norte, Natal 59078-900, Brazil

**Keywords:** drinking water, potable water, public health surveillance, quality control, government

## Abstract

This study identified and mapped worldwide surveillance actions and initiatives of drinking water quality implemented by government agencies and public health services. The scoping review was conducted between July 2021 and August 2022 based on the Joanna Briggs Institute method. The search was performed in relevant databases and gray literature; 49 studies were retrieved. Quantitative variables were presented as absolute and relative frequencies, while qualitative variables were analyzed using the IRaMuTeQ software. The actions developed worldwide and their impacts and results generated four thematic classes: (1) assessment of coverage, accessibility, quantity, and drinking water quality in routine and emergency situations; (2) analysis of physical–chemical and microbiological parameters in public supply networks or alternative water supply solutions; (3) identification of household water contamination, communication, and education with the community; (4) and investigation of water-borne disease outbreaks. Preliminary results were shared with stakeholders to favor knowledge dissemination.

## 1. Introduction

Water is the central element for sustainable development and essential to human survival, socioeconomic development, energy, and food production. Thus, a balance between human demands and commercial needs competing for the use of scarce water resources is needed, given the worldwide population growth [1,2,3].

Drinking water and sanitation are fundamental rights recognized by the United Nations (UN) since 2010 [2,4,5]. They also represent one of the main milestones for ensuring sufficient, safe, acceptable, and affordable water access, reducing diseases, and improving worldwide health, education, and economic productivity. Recently, access to water was included in the Sustainable Development Goals (SDG), which prioritize access to water for all by 2030 to reduce social and health inequalities [6]. However, this goal still needs to be achieved, considering the challenge of extending services to rural and poor areas with vulnerable populations [7]. Water scarcity, including limited access to safe drinking water and sanitation, the increased pressure on water resources and ecosystems, and the exacerbated risk of droughts and floods led the UN to create the Water Action Decade (2018–2028) to focus efforts to reach SDG 6 [1].

The direct use and contact with poor-quality water may harm the health and well-being of people, increasing the need for monitoring water quality. Moreover, low-income countries often do not conduct routine programs, leading to a lack or scarcity of water quality data and exposing the health of millions of people [3,8]. For this reason, the World Health Organization (WHO) has concentrated efforts on water safety since 1958 by publishing international standards and guidelines on drinking water quality [3].

However, more than these legal standards are required to ensure compliance and implementation, especially in regions with limited government and resources [9]. For example, a study illustrated the contamination of water sources by coliforms, bacterial or chemical contaminants, and high concentrations of arsenic, manganese, and iron [10]. Another study assessed drinking water quality in low- and middle-income countries and demonstrated that most of the studied population was exposed to fecal contamination; significant differences in water quality were also found between people from low- and middle-income countries [11]. These findings highlight the differences in quality between points of collection and use and the efforts needed to strengthen water quality surveillance by regulatory agencies [11]. 

Therefore, the surveillance of drinking water quality developed by public health authorities is important for public health protection because it identifies and assesses potential risks for collective health related to drinking water and contributes to improving water quality, quantity, accessibility, and coverage. This surveillance must also generate periodic reports about water quality and public health concerns and priorities, which may require exchanging information from various agencies [3]. 

Considering the need to establish surveillance actions of drinking water quality, it is essential to (1) identify and compare the actions developed worldwide to define gaps for future research and (2) understand how surveillance actions of drinking water quality are implemented by government agencies or public health services worldwide, including their impacts, results, challenges, and limitations.

Therefore, this study aimed to identify and map the surveillance actions and initiatives of drinking water quality implemented worldwide by governmental agencies and public health services.

## 2. Materials and Methods

### 2.1. Study Type

We conducted a scoping review between July 2021 and August 2022. This research method seeks to answer broad questions about the nature and diversity of the available evidence and provide an in-depth view of the topic [12,13]. All procedures were performed according to the Joanna Briggs Institute (JBI) [12] method, the theoretical framework proposed by Arksey and O’Malley [14], and updates by Levac, Colquhoun, and O’Brien, [15] and Peters et al. [12]. The following stages were used for elaborating the review: definition of the research question and inclusion criteria; description of study selection; evidence search; evidence selection; evidence extraction; evidence analysis; presentation of results; and summary of the evidence, conclusions, and implication of findings [12,15,16]. The results are presented following the Preferred Reporting Items for Systematic Reviews and Meta-analyses—Extension for scoping reviews (PRISMA-ScR) checklist [17].

The detailed methodology was published in a protocol [18]. The main alteration of this review was related to the mapping of studies: we used the world map from an electronic spreadsheet (Google Sheets, Google LLC, Mountain View, CA, USA) to better address the mapping, instead of the GeoDa software, 1.20 version (Center for Spatial Data Science, Chicago, IL, USA).

### 2.2. Definition of Research Questions and Inclusion Criteria

The following research questions were defined considering the mnemonic population, concept, and context after consensus among authors: (1) what are the worldwide actions and initiatives of surveillance of drinking water quality implemented by government agencies and public health services?; (2) what are the impacts and results of the implemented surveillance actions of drinking water quality?; and (3) what are the challenges and limitations for developing surveillance actions of drinking water quality?

We included publications—i.e., research articles (experience reports and quantitative, qualitative, or mixed studies), theses, dissertations, or official documents—regarding surveillance actions of drinking water quality implemented by public health agencies and authorities worldwide. No language, period, or country filters were considered. Duplicated studies, literature reviews, theoretical essays, editorials, expert opinions, conference and congress annals, and publications considering water for use other than human consumption were excluded. 

### 2.3. Description, Search, Selection, and Extraction of Evidence 

The search was conducted from 4 September 2021 to 29 November 2021, in three stages: (1) consultation of descriptors, keywords, and synonyms to elaborate the search, which was revised and improved by a librarian; (2) search in peer-reviewed (Medline/PubMed, Scopus, Web of Science, Embase, LILACS via Biblioteca Virtual em Saúde (BVS), and Engineering Village databases) and gray literature (Google Academic, Open Access Theses and Dissertations (OATD), ProQuest Dissertations & Theses Global (PQDT), Biblioteca Digital de Teses e Dissertações (BDTD), and Catálogo de Teses & Dissertações—CAPES); and (3) manual consultation in the reference list of selected publications. 

Study selection followed the steps of identification, screening, eligibility, and inclusion [19]. The Rayyan [20] software free version (Qatar foundation, Doha, Qatar) was used to organize the results, remove duplicates, and analyze titles and abstracts. The remaining stages of the review were blindly conducted by two authors (RHL and CRDVS). These authors also performed a pilot test in each study stage to calibrate and adapt the protocol. All conflicts were resolved by consensus. 

Two independent reviewers (RHL and CRDVS) extracted the evidence. The form was based on the JBI Manual for Evidence Synthesis [12] and included the following variables: type of material, year of publication, publication context, academic degree of the main author, aims, type of research, surveillance actions and the responsible agencies, impacts or results, and challenges or limitations. The peer extraction and ethical attention of researchers ensured the reliability and consistency of the extracted data.

### 2.4. Analysis of Evidence

A descriptive synthesis was performed, and quantitative variables (i.e., type of material, year, context, academic degree, research, and agency/professional responsible for the action) were presented as absolute and relative frequencies. Regarding qualitative variables (i.e., surveillance action, impacts or results, and challenges and limitations of the actions), textual fragments extracted from results and conclusions of the included publications were analyzed using the Interface de R pour les Analyses Multidimensionnelles de Textes et de Questionnaires (IRaMuTeQ) software version 0.7 alpha 2. This software analyzes the structure and text organization and informs connections between the most frequently reported lexical worlds [21,22]. All textual fragments were translated into Portuguese (i.e., native language of the researchers) and revised to produce the corpus and subsequent analyses. 

The variables related to surveillance of drinking water quality conducted worldwide and its impacts and results were analyzed using descending hierarchical classification. This analysis classifies text segments according to their vocabularies and frequency of lemmatized words to obtain classes of elementary context units composed of vocabularies similar to each other but different from other classes [21,22]. The emerging classes were identified according to a dendrogram elaborated using chi-square (χ^2^) values. High χ^2^ values represented strong associations between word and class; words with χ^2^ < 3.84 were not considered (*p* < 0.05). The challenges and limitations of surveillance actions were submitted to a similarity analysis that allows the identification of co-occurrences between words and indicates connections that help identify the structure of a corpus [21,22].

### 2.5. Consultation with Stakeholders

This stage was structured to improve the strength of the review and favor socialization and knowledge transfer to those interested in the area [15]. A summary of results containing the file and link to a Google form was sent via email to stakeholders, who were selected according to their expertise in monitoring drinking water quality or publications in the area.

Thus, preliminary results were analyzed by seven specialists in surveillance of drinking water quality. The following questions were considered [15]: ways of disseminating knowledge, applicability of services in water quality surveillance, and research gaps or suggestions for future studies. This stage was approved by the research ethics committee, and all stakeholders signed an informed consent form. Other details were mentioned in the study protocol [18].

## 3. Results and Discussion

The identification process resulted in 26,235 studies, of which 4388 were from searches in databases (Scopus: 2030; Medline/PubMed: 720; Engineering Village: 676; Web of Science: 546; EMBASE: 289; LILACS via BVS—Portuguese: 94; and LILACS via BVS—English: 33) and 21,847 from the gray literature (Google Scholar: 21,800; Catálogo de Teses e Dissertações—CAPES: 36; PQDT: 7; BDTD: 2; and OATD: 2). The flowchart of study selection shown in Figure 1 presents all reasons that led to the exclusion of publications in each phase of the study. We emphasize that this selection was conducted independently by two authors. 

A total of 47 publications were selected after the screening and eligibility stages, and two references were added after consultation on reference lists, resulting in a final sample of 49 studies (Appendix A).

### 3.1. Characterization of Included Studies 

The most included studies were peer-reviewed articles (32; 65.3%), followed by reports of surveillance actions (13; 26.5%) and dissertations of Master’s in Science (4; 8.2%). The number of publications increased between 1993 and 2020, mainly in 2017 (6; 12.2%). According to studies, surveillance actions were developed in several countries, mostly in the USA (17; 34.7%) and Brazil (14; 28.6%) (Figure 2).

The USA has assumed the leading role in developing concepts and models of public health surveillance. The Center for Diseases Control, for example, was a catalyst for this process and is responsible for public health surveillance and outbreaks of water-borne diseases (WBDOS) in the country; the latter is currently a publication of the National Notifiable Diseases Surveillance System [23,24]. In addition, the United States Environmental Protection Agency approved the Safe Drinking Water Act in 1974 as a form of public health protection [25]. 

Brazil stands out in the surveillance of drinking water quality for human consumption by assigning the Unified Health System (SUS) to perform water inspections and elaborate policies and actions related to basic sanitation [26]. Furthermore, specific legislation supports the control, surveillance, and drinking water standards for human consumption [27]. The SUS also has the “Vigiagua” and “Sisagua” programs, which provide public domain data for research, action management, and monitoring of drinking water quality [28]. Brazil is one of the few countries that routinely publish data favoring an in-depth analysis of inequalities in drinking water quality [7].

The career of the first author was identified in 40 studies (81.6%); most were medical doctors (6; 12.2%), nurses (5; 10.2%), biological and chemical scientists (4; 8.2%), civil engineers, and veterinarians (3; 6.1%). Other less-identified careers were in health, engineering, environmental, and social areas. The various careers reflect the need for action in the area and the articulation of different knowledge and practices [3]. 

The study design was described in some publications (24; 48.90%): descriptive (8; 16.32%) [29,30,31,32,33,34,35,36], cross-sectional or ecological (4; 8.16%) [37,38,39,40], documentary (3; 6.12%) [41,42,43], retrospective cohort [44,45], epidemiological [46,47], case study [48], action research [49], experience report [50], and quantitative–qualitative evaluative and mixed-methods research [51]. Other publications were reports regarding surveillance actions (14; 28.57%) [52,53,54,55,56,57,58,59,60,61,62,63,64,65]. Reports describing water quality must be produced periodically by public health authorities, integrated into management and policy actions, and combined with better dissemination and communication among stakeholders [3,8]. 

Those responsible for health surveillance actions were professionals linked to public health departments (40; 81.63%) from municipal, state, or federal spheres [30,31,32,33,34,35,36,37,38,39,41,42,45,46,47,48,49,50,52,53,54,55,56,57,58,59,60,61,62,64,66,67,68,69,70,71,72,73,74,75]; agencies responsible for water protection [44,76]; departments of environment and conservation [40]; departments of public health engineering [77]; national water laboratories [29]; or in-field sanitary inspectors of a water test laboratory [51]. Sometimes, the surveillance involved other sectors besides health (e.g., environmental, civil protection, and agencies responsible for the supply) [44,45,46,48,54,64] or was conducted by a national network for monitoring environmental public health [63,65].

The superposition of several ministries and organizations responsible for water management hinders the management of this area, which requires an institutional approach to provide quality since it is essential to human development [8]. However, the reality observed in the studies showed that most ministries of health (or related to public health) were responsible for the surveillance of drinking water; in other countries, an agency related to environmental protection was responsible for this initiative [3].

### 3.2. Surveillance Actions of Drinking Water Quality and Their Impacts and Results: Descending Hierarchical Classification 

The corpus was composed of 49 texts divided into 207 segments of text (ST), of which 176 ST were used (85.02%). A total of 7563 occurrences were found (words, shapes, or vocables): 1640 were distinct words, and 911 occurred only once. The analyzed content was categorized into four classes and named according to Figure 3.

#### 3.2.1. Class 1—Assessment of Coverage, Accessibility, Quantity, and Drinking Water Quality in Routine and Emergency Situations 

This class represented the actions developed by authorities responsible for monitoring drinking water quality in routine and emergency situations. The actions sought to ensure adequate drinking water quality, quantity, and access (supply systems) for the population. In most cases, these actions required the articulation between services responsible for water supply and other services responsible for public health surveillance. 

A total of 20.4% of the analyzed material was represented in this class (f = 36 ST), composed of words and radicals (e.g., analysis, quality, surveillance, technical, coverage, and consumption) in the interval between χ^2^ = 3.91 (health) and χ^2^ = 31.57 (action).

Surveillance actions assessing the coverage, quantity, and accessibility of drinking water for the population were poorly highlighted in studies conducted in Costa Rica [29,43], Brazil [34], and India [51]. Some studies revealed locations in which almost the entire population was supplied [29,43]. Other studies demonstrated that, although all small residences had access to water, the offer was limited to two hours per day, motivating water storage and use of pumps to obtain more water during the supply period [51], and verifying the consumption per capita was impossible due to limitations in the information system [34].

In this sense, the literature indicates that the operational and maintenance costs for expanding drinking water and sanitation services hindered many countries from adequately providing these services to the population. About 2 billion people do not have access to drinking water, reflecting the worldwide crisis caused by two major structural flaws: inequality and poverty. The former results from lack of support from socioeconomic systems, while the latter is related to the unsustainable relation with aquatic ecosystems, which transforms water into a vector of disease and death [78].

Most studies included in this class highlighted actions related to assessing water quality offered in routine situations according to contaminant levels [29,43,69,71,74,76]. They also mentioned a possible seasonal pattern in water contamination during winter and summer [69] or in March and April [34].

It is known that changes in rainfall impact water quality (i.e., increased rainfalls may overload domestic and industrial sewage collection systems), leading to the disposal of raw sewage into water courses and increasing the risk of contamination by pathogens. Alternatively, droughts may increase the salinity of the soil and freshwater sources, decreasing the dilution of pollutants [8] and, consequently, groundwater formation and water supply for part of the population [79]. Thus, climate variables (e.g., precipitation, humidity, evaporation, and temperature) may contribute to outbreaks of diarrheal diseases [80], which may mostly affect children below five during very dry, hot, humid seasons [81].

Another important result was the difference in data and information related to water quality between agencies responsible for water supply and health surveillance [37,41]. This situation is problematic since data are crucial to policy-making, development of new programs and interventions, and improvement of public communication, research assessment, and investment allocation [82]. Additionally, divergences may compromise the correct alignment of adequate interventions.

Nevertheless, the availability of global indicators related to water supply, sanitation, and hygiene constantly improved in the first five years of the SDGs. Moreover, national estimations related to the safe management of drinking water increased from 96 to 138 countries, including in rural (from 20 to 65 countries) and urban regions (from 42 to 87 countries) [7].

Concerns about water contaminants must also be highlighted. Currently, thermotolerant coliforms are the most reported water contaminants; however, contamination by hydrocarbons and pesticides from local industrial and agriculture models is also worrisome [29]. For example, a study in Denmark found that only 0.5% of samples from public drinking water contained pesticides, and 16% of these exceeded the recommended levels under the national standard [83]. This reality is very different from that in low-income countries, where agriculture and raw sewage threaten water quality [8]. 

Surveillance actions of drinking water quality in emergency situations were related to natural or anthropogenic disasters, such as tsunamis [48], dam failure accidents [50], and environmental disasters in the paper industry [30]. The actions involved the identification of priorities related to drinking water quality [48], including periodic monitoring and collective and individual supply solutions to gather water from underground sources in the affected municipalities [50]; detection of vulnerable points in the network [30] to improve the microbiological and chemical safety of water; and analysis of household wells to classify the risk in water consumption [9]. The need to create intersectoral technical working groups [30] or operating committees for integrating essential areas to attend to emergencies [50] was also highlighted, besides the installation of chlorinated water tanks; distribution of domestic water treatment reagents to residences in the affected areas [64]; and improvements in water management related to source protection, disinfection practices, and attention to contamination [48].

The increased exposure to risks caused by consumption patterns, working conditions, and exposure to chemicals associated with the appropriation of nature by humankind is leading to environmental degradation, which may increase the occurrence of disasters and emergencies in public health. This implicates the need for immediate action not only in health care and surveillance but also in other areas of action according to the characteristics of the event [84].

In this context, the effects of natural disasters on health represented an important challenge for public health due to several derived factors (e.g., vulnerability of the population and the economic development model adopted, which often affects the climate) and required coordinated risk management action among all levels across health sectors [85].

#### 3.2.2. Class 2—Analysis of Physical–Chemical and Microbiological Parameters in Public Supply Networks and Alternative Water Supply Solutions 

Class 2 was at the same level as class 1 (Figure 3). Class 2 was based on the assessment of parameters related to the supply of quality water and linked to specific legislation in countries that standardized and limited the contaminant levels for coliforms, physical–chemical parameters (e.g., turbidity, pH, color, and temperature), and other microbiological indicators. The essential role of routine and systematized information regarding these parameters was highlighted to support health surveillance actions.

Class 2 comprised 45 ST (25.60%), with words between χ^2^ = 4.48 (total coliforms) and χ^2^ = 39.09 (sample). The following words were presented: residual free chlorine (χ^2^ = 28.72), parameter (χ^2^ = 22.60), turbidity (χ^2^ = 14.55), thermotolerant coliforms (χ^2^ = 11.92), *Escherichia Coli* (χ^2^ = 11.43), improper (χ^2^ = 10.89), and legislation (χ^2^ = 5.26).

Residual-free chlorine is a widely searched indicator used to assess water quality [32,34,35,37,39,41,51,64]. Inadequate chlorination was identified as a vulnerable factor in alternative solutions for water supply, exposing the population to microbiological contamination [32]. In addition, turbidity was analyzed as a relevant indicator [34,35,37,39,40,41,49,51,53] and frequently associated with acute diarrhea when the maximum limit was exceeded [37]. Bacterial contamination by coliform [32,33,34,37,39,41,45,51,68,72] and *Escherichia coli* [33,35,39,45,51,64,75] were also correlated with the emergence of acute diarrhea [75]. The assessment of fluoride levels was indicated as a potential parameter for preventing cavities in the population [31,35,36,41,47].

Similar to our findings, a study in Zimbabwe investigated the quality of alternative water sources and their distribution system. The authors revealed that, although samples met the WHO recommendations for physical–chemical parameters, the microbiological requisites were not met due to coliform contamination regardless of water source and location. Moreover, *Escherichia coli* and *Salmonella spp*. were found in some samples. Multidrug resistance (amoxicillin, ampicillin, and cephalothin) was also identified, indicating that water without further treatment was unsafe for human consumption [86].

Other problems identified by studies included in this review were the need for more information or irregular registration of some parameters [35,75], lack of inspection of water quality and registration of water sources [34], and reduced compliance with surveillance actions in rural areas [39]. In this sense, it is essential to use open-access databases to centralize the register of analyses and allow the assessment of drinking water using a spatial–temporal categorization [83]. 

#### 3.2.3. Class 3—Identification of Household Water Contamination, Communication, and Education with the Community

This class was linked to Classes 1 and 2 on the external level of the dendrogram and represented the identification of water contaminants at the household level and dissemination and education needed with the community, especially in situations of risk.

Class 3 comprised 33% of ST analyzed (f = 58 ST) and considered words in the interval between χ^2^ = 16.96 (use) and χ^2^ = 4.88 (affected). The following words were presented: well (χ^2^ = 13.99), family (χ^2^ = 13.43), community (χ^2^ = 11.29), drinking (χ^2^ = 11.29), residence (χ^2^ = 10.47), counseling (χ^2^ = 7.14), warning (χ^2^ = 6.21), domestic (χ^2^ = 6.21), communication (χ^2^ = 5.16), and domicile (χ^2^ = 5.16).

The surveillance actions of water quality highlighted in this class addressed the identification of chemical contaminants in households (i.e., lead [70,72], arsenic [66,71,72], iron [70,72,77], nitrate [68,72,77], fluoride [46,77], aluminum, manganese, strontium, and nitrogen [72]), revealed locations that did not fully comply with regulations of chemical contamination [48], and identified volatile organic compounds within the allowed level [68]. Identifying these contaminants has been part of testing programs at residences of people from at-risk groups (e.g., women with children [70], low-income families with pregnant women, and young children [72]) and assessments of supply sources and sources needing treatment after a disaster [50]. 

Chemical contamination represents a major challenge for public health because of its effects after long-term exposures, interfering with the comprehension of risky situations and elaboration of preventive measures. Furthermore, emerging pollutants (e.g., medicines, endocrine deregulators, pesticides, organic products, metals, and illicit drugs) are not routinely monitored because they are not listed as common chemical contaminants [87]. In this sense, efforts are needed to regulate the monitoring of these substances, considering the economic activities or local characteristics of sanitary sewage [87]. For instance, the extensive use of nitrogen-containing fertilizers raises nitrate levels in groundwater and increases the risk for people consuming this water; thus, the protection of groundwater source areas is needed to meet the supply needs of economic development [88].

For this reason, it is important to assess the quality of drinking water, especially regarding household chemical levels, since the availability of drinking water is at risk due to natural and anthropogenic activities. A study conducted in a rural region of Bangladesh observed that most families consumed drinking water of poor quality and containing high levels of iron, manganese, and arsenic [89].

The following methods of education and communication with the community were developed in situations of microcystin contamination: alerts to the population (e.g., warnings to boil water) [62], risk classification of wells (including with *Salmonella spp*.) [48], guidance for families affected by disasters [50], counseling in residential and commercial systems [38], and conscientization campaigns for the rural population to improve water sources and comprehend the role of drinking water in health [77].

In this context, risk communication by professionals working with drinking water may occur on many occasions, such as unexpected events or disasters (e.g., chemical spills, outbreaks, hurricanes, and power outages) or during a routine inspection [90]. Water systems or agencies must provide information to the population, encourage action preparation and recommendations, or comply with precepts of public legal notification when its quality is compromised. These warnings must include information about water boiling and avoiding its use for drinking or other purposes [91].

The risks from environmental issues may be seen as more alarming and be less understood than other health risks because they are often invisible and disproportionately affect part of the population; thus, requiring communication professionals with the ability to explain situations clearly, succinctly, and empathically [90].

The effectiveness of risk communication relies on health literacy issues, which represent the ability to receive information, comprehend, and act adequately to make decisions [91]. Furthermore, recognizing the characteristics of the target audience (e.g., age, socioeconomic level, educational level, culture, language, environmental preoccupation, and environmental health literacy) is essential for effective and transparent communication [92]. 

The recognition of work, continuing education of workers, community participation, social responsibility, social accountability, political support, and personal commitment were also reported as essential elements for successful surveillance actions of water quality [49]. Furthermore, the training of health workers is needed, especially in low-income countries, to improve the use of geographical information tools and increase the interconnection between human and environmental health, socioeconomic transitions, and climate changes. Data from passive surveillance, disease outbreak reporting systems, and environmental and climatic observations may also be used to assess patterns and trends of diseases [93].

Public health surveillance systems must be strengthened by adequately allocating resources and constantly training to use information in the prioritization, planning, action, and assessment of actions [23]. Furthermore, the participation of citizens in monitoring population health should be explored [93].

#### 3.2.4. Class 4—Investigation of Water-Borne Disease Outbreaks

Class 4 comprised the most different topic and represented the investigation of factors associated with WBDOS, including data analysis regarding drinking water affected by outbreaks that caused illness, hospitalization, and death [53,56,57,58,59,60,61,62,68].

This class comprised 21% of the material analyzed (f = 37 ST) and considered words in the interval between χ^2^ = 124.60 (outbreak) and χ^2^ = 4.24 (positive). It presented representative words, such as associated (χ^2^ = 64.98), WBDOS (χ^2^ = 44.08), transmitting (χ^2^ = 38.71), disease (χ^2^ = 36.49), death (χ^2^ = 35.63), etiology (χ^2^ = 31.49), investigation (χ^2^ = 22.60), chemical (χ^2^ = 22.21), infectious (χ^2^ = 19.33), intoxication (χ^2^ = 15.38), virus (χ^2^ = 11.47), parasite (χ^2^ = 10.78), and bacteria (χ^2^ = 5.34).

Several states of the USA and Canada investigated reported or suspected cases of WBDOS, in which etiological agents were unknown [67], unidentified [52,56], or linked to infectious agents (i.e., bacteria, viruses, and parasites) [55,56,57,58,59,60,61,67]; some cases were related to multiple etiology [60,61]. Chemical poisoning by sodium hydroxide [55,57,61], copper [55,56,58], nitrate [52,57], nitrite [55], tilbenzene, toluene, xylene [58], and toxins [59] was also reported.

Outbreaks in the USA and Canada were associated with the community, non-community, or individual water systems [55,56] and unregulated private or non-community wells [57]; nearly half of the outbreaks investigated were in semi-public systems [67]. Superficial [56] and groundwater [52,57,60] sources of these systems were also linked to outbreaks.

In the USA, the public health authority uses data from the states to monitor diseases, risk factors, and public health issues of collective interest. They also screen for outbreaks related to contaminated water, such as *Escherichia coli* or other non-infectious causes (e.g., lead poisoning) [82], to reveal deficiencies in the water supply system.

Deficiencies possibly associated with the emergence of outbreaks were linked to a water source, treatment plant or distribution system [59,60,61], and use of non-treated groundwater [61]; some were unknown or out of the jurisdiction of water utility or location of use [59,60]. Gastrointestinal diseases [46,52], acute diarrhea [53], hepatitis A, cryptosporidiosis, and giardiasis [52] were also present, and more serious cases resulted in death (e.g., chemical poisoning by fluoride) [52].

### 3.3. Challenges and Limitations of Surveillance Actions of Water Quality: Similarity Analysis

The corpus produced by the variable “challenges and limitations of surveillance actions of water quality” comprised 49 texts divided into 76 ST; 54 ST (71.05%) were used. A total of 2537 occurrences emerged (words, forms, and vocables): 872 were different words, and 596 had a single occurrence. The minimum cutoff point for similarity analysis was twice the ratio between the number of occurrences and the number of forms [22,23]. In this case, the frequency considered for the analysis was 3 × 2.9 = 8.7 of words above this cutoff and presented theoretical coherence with the analyzed results, except for the words “no”, “as”, and “more”. 

The material analyzed produced an image comprising five halo communities; three words that stood out: “water”, “data”, and “surveillance”. The former is the central element derived from other communities and words. The central part was composed of the word “water” linked to “potable”, “quality”, “information”, and “lack”, ramifications of which were composed of the following words: “treatment”, “system”, “analysis”, and “municipalities” (right ramification); “source”, “population”, and “difficulty” (central ramification); and “notification”, “outbreak”, “disease”, “location”, “health”, and “data” (left ramification). A sub-ramification derived from the left ramification was composed of the words “report”, “number”, and “sample” (Figure 4).

The main challenges and limitations related to surveillance actions of water quality were: (1) lack of information about drinking water quality due to insufficient samples; (2) lack of data standardization regarding the analysis of systems supply provided by municipalities; and (3) difficulties in identifying sources and water treatment, which hindered the surveillance and report of outbreaks associated with water-borne diseases.

In this sense, public health surveillance actions still lack information systems that aggregate relevant and essential data from different sources. Although the coverage of surveillance systems is still limited, fragmented, and unequal, and hinders knowledge exchange and integration, computational technologies may help optimize their quality, capacity, and efficacy [23].

The monitoring and assessment of water quality offer important opportunities for innovation in the 21st century, especially regarding information and communication technology, that may contribute to several strategies (e.g., use of data for modeling, remote sensing, and machine-learning approaches). These technologies and science initiatives may fill data gaps [94,95] and contribute to achieving SDG [8,93].

### 3.4. Consultation with Stakeholders 

Seven stakeholders (five technical professionals in the area and two researchers) with practical or research experience in water quality surveillance for human consumption responded to the form requesting an appraisal of the results.

Regarding the disclosure of results, the publication of a scientific article was almost unanimous. One respondent indicated an open publication (e.g., booklet or newspaper) for easy access to the population, while another indicated the presentation of results to those responsible for monitoring water quality in municipalities. Stakeholders also discussed the applicability of results, which was considered support for action. For example, results could help identify the main challenges to action development and potential adaptations, guide activities, and improve and qualify surveillance actions of water quality for human consumption in the countries.

Gaps and suggestions for future studies comprised the need for (1) greater clarity on the concept of monitoring drinking water quality; (2) on-site visits for the appropriation of developed actions; (3) conceptualization of didactic material for the population; (4) assessment of water quality in areas of informal occupation, flooding, or use of pesticides; (5) assessment of water quality indicators and WBDOS, (6) assessment of the impacts of actions on the community, (7) incorporation of information on water security plans, and (8) use of information systems by the countries.

### 3.5. Limitations

Despite the highly sensitive search strategy, important documents from countries may have not been retrieved during searches in databases. Although we did not limit the language for inclusion of studies, the search performed with terms only in English and Portuguese may be considered a limitation of the study.

Another limitation of this study was the inclusion of publications that portrayed surveillance actions and initiatives of water quality for human consumption developed only by the government and public health agencies. This may have excluded countries that use other types of services for this function. Nevertheless, this criterion was adopted as a contextual approach in our study.

## 4. Conclusions

This scoping review identified and mapped the available scientific evidence on the main worldwide actions and initiatives related to the surveillance of drinking water quality. The results revealed (1) several countries that published actions developed by professionals with varied academic backgrounds; (2) those with responsibility for health surveillance actions were professionals linked to public health departments or agencies responsible for water protection; and surveillance reports were the most identified publications. The four thematic classes in this review detailed the actions and their results in the countries. The preliminary results were also shared with stakeholders and favored early knowledge dissemination.

This review identified important issues that need to be analyzed so that surveillance actions of drinking water quality contribute to safe access to water, such as the qualification of professionals for routine and emergency situations, including those with the ability to perform risk communication; the capacity to consider the health literacy of populations to achieve better results in actions related to health information, communication, and education; the need for surveillance systems of water quality responsive to outbreaks or water-related issues affecting human health; and the reduction of inequalities related to access to quality water between rural and urban areas. These points also represent research gaps that must be addressed in future studies to improve surveillance actions of drinking water quality.

## Figures and Tables

**Figure 1 ijerph-20-00559-f001:**
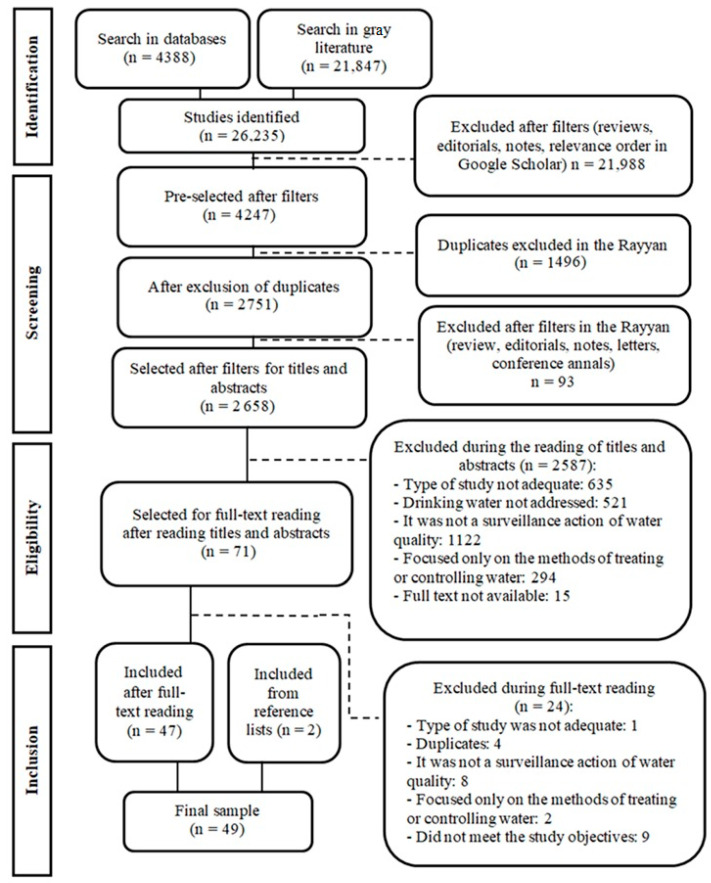
Flowchart of study selection adapted from the Preferred Reporting Items for Systematic Review and Meta-Analyzes [20].

**Figure 2 ijerph-20-00559-f002:**
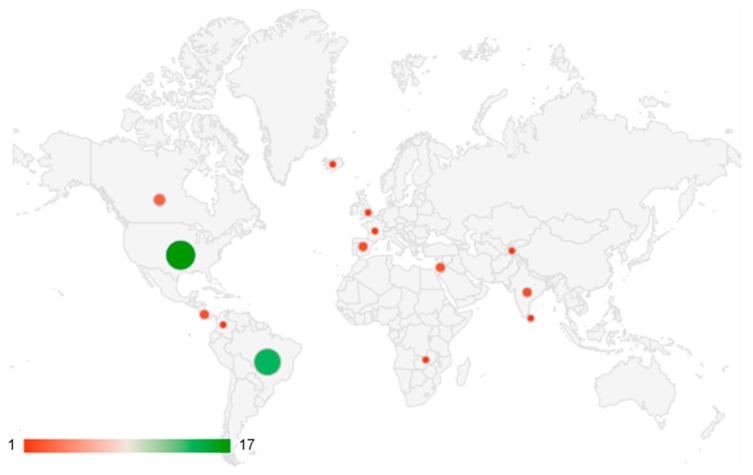
Distribution of surveillance actions developed worldwide identified in the review.

**Figure 3 ijerph-20-00559-f003:**
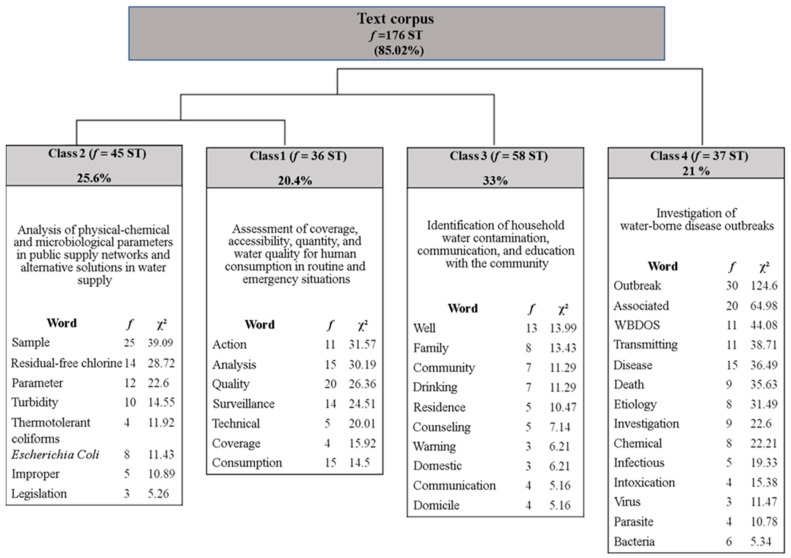
Dendrogram of the descending hierarchical classification describing classes, frequency, and the chi-square (χ^2^). Legend: ST—segments of text; f—frequency; χ^2^—chi-square; WBDOS—water-borne diseases.

**Figure 4 ijerph-20-00559-f004:**
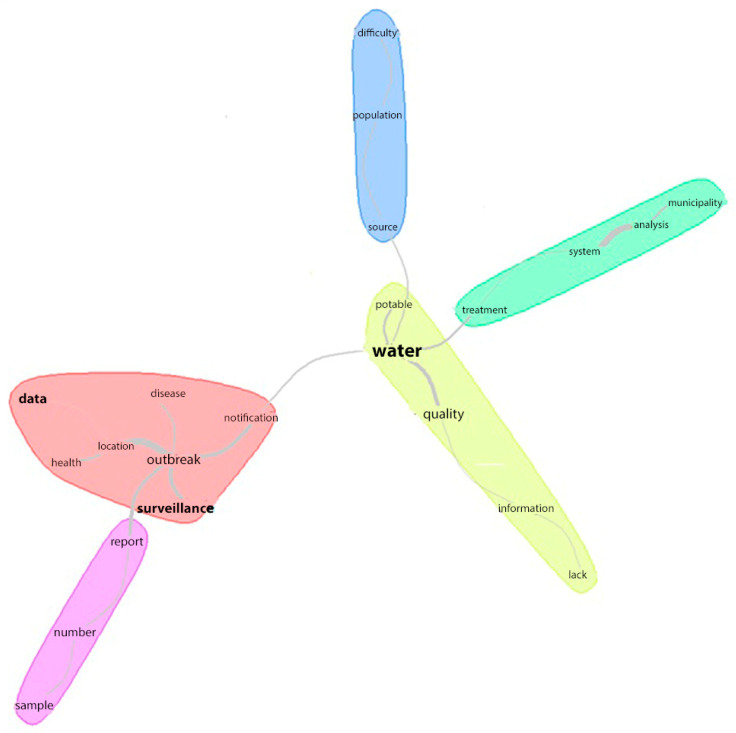
Similarity analysis of challenges and limitations of surveillance of drinking water quality worldwide. Presentation Fruchterman Reingold, chi-square scores with community and halo.

## Data Availability

Not applicable.

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
