# Peer review of "Worldwide Surveillance Actions and Initiatives of Drinking Water Quality: A Scoping Review"

_ijerph, 2022, doi:10.3390/ijerph20010559_

Round 1

Reviewer 1 Report (Previous Reviewer 2)

Accept the correction to my review comments. 

This manuscript is a resubmission of an earlier submission. The following is a list of the peer review reports and author responses from that submission.

Round 1

Reviewer 1 Report

Reviewer comment on the manuscript numbered 1941234 titled “Worldwide Surveillance Actions and Initiatives of Drinking Water Quality: A Scoping Review” submitted to the International Journal of Environmental Research and Public Health.

Comments:

Water quality is a global issues and various initiatives have been taken to ensure safe sources of drinking water to all. However, water is becoming an important and challenging issue not only in the terms of quantity but in the quality aspects. The authors have raised an important aspects to study the scoping review on drinking water quality and the surveillance and initiatives on these issues at the global levels. However, some of the issues needs to ascertain before it may be recommended for the acceptance of the manuscript to the IJERPH. Some of my specific comments are:

1. At the end of the introduction, please add the outline of the paper.

2. Figure 1 is well explained. My only concern is how the authors have excluded is not clear to me. Please explain it in the text.

3. Figure 2 is not clear. I hope the authors have tried to show the distributions of surveillance actions developed at global level. However, I am unable to understand not to see many countries not to be part in the Figure. Please recheck and validate it again if possible.

4. Figure 3 is pretty clear. However, authors should discuss in 1-2 line about Descending Hierarchical Classification in the text.

5. Section 2 on Materials and Methods can be improved. For example, 2.1 titled study type can be improved and section 2.5 titled consultation with stakeholders is not clearly coming in the current form. Please elaborate on these.

6. The authors should shift limitations (section 3.5) in the Conclusions section.

7. Please changed the conclusions part. It should be the summary on what you have done in the manuscript rather than a general summary.

8. I feel authors should include some of the studies which are really important when it comes to study heavy metal contamination in drinking water. Some of literature I am as an example:

https://doi.org/10.1016/j.gsd.2022.100735

https://doi.org/10.1016/j.gsd.2020.100504

Akter, T., Jhohura, F. T., Akter, F., Chowdhury, T. R., Mistry, S. K., Dey, D., ... & Rahman, M. (2016). Water Quality Index for measuring drinking water quality in rural Bangladesh: a cross-sectional study. Journal of Health, Population and Nutrition35(1), 1-12.

Bonansea, M., Rodriguez, M. C., Pinotti, L., & Ferrero, S. (2015). Using multi-temporal Landsat imagery and linear mixed models for assessing water quality parameters in Río Tercero reservoir (Argentina). Remote Sensing of Environment158, 28-41.

Rigacci, L. N., Giorgi, A. D., Vilches, C. S., Ossana, N. A., & Salibián, A. (2013). Effect of a reservoir in the water quality of the Reconquista River, Buenos Aires, Argentina. Environmental monitoring and assessment185(11), 9161-9168.

Lopes, F. B., Andrade, E. M. D., Meireles, A., Becker, H., & Batista, A. A. (2014). Assessment of the water quality in a large reservoir in semiarid region of Brazil. Revista Brasileira de Engenharia Agrícola e Ambiental18, 437-445.

9. The manuscript needs language editing as I found throught the problem in English.

I congratulate authors for an interesting work to carry out. Wish them a good luck. 

Reviewer 2 Report

I have the following comments: 

I find reading the paper confusing, as the same topic appear in all classes, as chemical contamination.   

Line 255-256: It is stated the operational and maintenance cost from infrasturcure is a hindrance but it is not part of the Class 1 words.

Line 353: add in Denmark  (Is Denmark missing on Figure 2)

Figure 4:  use large font size. 

Line 522:  What were the important issues?

The conclusion chapter is not very clear and needs to be re-written. I do not see the reason for the conclusion on USA and Brazil (line 516-517). Data does not provide for statisticly sound conclusion on this I would think.